# ROBOPIANIST: Dexterous Piano Playing with Deep Reinforcement Learning

**Kevin Zakka**[β, δ], **Philipp Wu**[β], **Laura Smith**[β],
**Nimrod Gileadi**[δ], **Taylor Howell**[σ], **Xue Bin Peng**[ψ], **Sumeet Singh**[δ], **Yuval Tassa**[δ],
**Pete Florence**[δ], **Andy Zeng**[δ], **Pieter Abbeel**[β]
[β]UC Berkeley, [δ]Google DeepMind, [σ]Stanford University, [ψ]Simon Fraser University
Correspondence to: `zakka@berkeley.edu`

**Abstract:** Replicating human-like dexterity in robot hands represents one of the largest open problems in robotics. Reinforcement learning is a promising approach that has achieved impressive progress in the last few years; however, the class of problems it has typically addressed corresponds to a rather narrow definition of dexterity as compared to human capabilities. To address this gap, we investigate piano-playing, a skill that challenges even the human limits of dexterity, as a means to test high-dimensional control, and which requires high spatial and temporal precision, and complex finger coordination and planning. We introduce ROBOPIANIST, a system that enables simulated anthropomorphic hands to learn an extensive repertoire of **150** piano pieces where traditional model-based optimization struggles. We additionally introduce an open-sourced environment, benchmark of tasks, interpretable evaluation metrics, and open challenges for future study. Our website featuring videos, code, and datasets is available at https://kzakka.com/robopianist/.

**Keywords:** high-dimensional control, bi-manual dexterity

## 1 Introduction

Despite decades-long research into replicating the dexterity of the human hand, high-dimensional control remains a grand challenge in robotics. This topic has inspired considerable research from both mechanical design [1, 2, 3] and control theoretic points of view [4, 5, 6, 7, 8]. Learning-based approaches have dominated the recent literature, demonstrating proficiency with in-hand cube orientation and manipulation [9, 10, 11] and have scaled to a wide variety of geometries [12, 13, 14, 15]. These tasks, however, correspond to a narrow set of dexterous behaviors relative to the breadth of human capabilities. In particular, most tasks are well-specified using a single goal state or termination condition, limiting the complexity of the solution space and often yielding unnatural-looking behaviors so long as they satisfy the goal state. How can we bestow robots with artificial embodied intelligence that exhibits the same precision and agility as the human motor control system?

In this work, we seek to challenge our methods with tasks commensurate with this complexity and with the goal of emergent human-like dexterous capabilities. To this end, we introduce a family of tasks where success exemplifies many of the properties that we seek in high-dimensional control policies. Our unique desiderata are (i) spatial and temporal precision, (ii) coordination, and (iii) planning. We thus built an anthropomorphic simulated robot system, consisting of two robot hands situated at a piano, whose goal is to play a variety of piano pieces, i.e., correctly pressing sequences of keys on a keyboard, conditioned on sheet music, in the form of a Musical Instrument Digital Interface (MIDI) transcription (see Figure 1). The robot hands exhibit high degrees of freedom (22 actuators per hand, for a total of 44), and are partially underactuated, akin to human hands. Controlling this system entails sequencing actions so that the hands are able to hit exactly the right notes at exactly the right times; simultaneously achieving multiple different goals, in this case, fingers on

7th Conference on Robot Learning (CoRL 2023), Atlanta, USA.

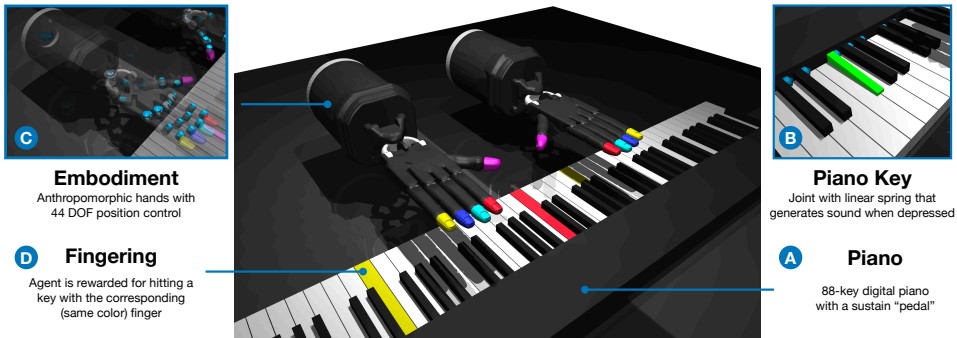

Figure 1: ROBOPIANIST simulation featuring a full-size digital keyboard (A) with 88 piano keys modeled as linear springs (B). In the piano playing task, two (left and right) anthropomorphic Shadow hands (C) are tasked with playing a musical piece encoded as a trajectory of key presses (D).

each hand hitting different notes without colliding; planning how to press keys in anticipation of how this would enable the hands to reach later notes under space and time constraints.

We propose ROBOPIANIST, an end-to-end system that leverages deep reinforcement learning (RL) to synthesize policies capable of playing a diverse repertoire of musical pieces on the piano. We show that a combination of careful system design and human priors (in the form of fingering annotations) is crucial to its performance. Furthermore, we introduce ROBOPIANIST-REPERTOIRE-150, a benchmark of 150 songs, which allows us to comprehensively evaluate our proposed system and show that it surpasses a strong model-based approach by over 83%. Finally, we demonstrate the effectiveness of multi-task imitation learning in training a single policy capable of playing multiple songs. To facilitate further research and provide a challenging benchmark for high-dimensional control, we open source the piano-playing environment along with ROBOPIANIST-REPERTOIRE-150 at https://kzakka.com/robopianist/.

## 2   Related Work

We address related work within two primary areas: dexterous high-dimensional control, and robotic pianists. For a more comprehensive related work, please see Appendix A.

**Dexterous Manipulation as a High-Dimensional Control Problem.** The vast majority of the manipulation literature uses lower-dimensional systems (i.e., single-arm, simple end-effectors), which circumvents challenges that arise in more complex systems. Specifically, only a handful of general-purpose policy optimization methods have been shown to work on high-dimensional hands, even for a single hand [10, 9, 12, 11, 16, 14, 17, 7], and of these, only a subset has demonstrated results in the real world [10, 9, 12, 11, 16]. Results with bi-manual hands are even rarer [15, 18]. In addition, the class of problems generally tackled in these settings corresponds to a definition of dexterity pertaining to traditional manipulation skills [19], such as re-orientation, relocation, manipulating simply-articulated objects (e.g., door-opening, ball throwing and catching), and using simple tools (e.g., hammer) [20, 21, 15, 11, 22, 12]. This effectively reduces the search space for controls to predominantly a single "basin-of-attraction" in behavior space per task. In contrast, our piano-playing task encompasses a more complex notion of a goal, extendable to arbitrary difficulty by only varying the musical score.

**Robotic Piano Playing.** Robotic pianists have a rich history within the literature, with several works dedicated to the design of specialized hardware [23, 24, 25, 26, 27, 28], and/or customized controllers for playing back a song using pre-programmed commands [29, 30]. The works of Scholz [31], Yeon [32] use a dexterous hand to play the piano by leveraging a combination of inverse kinematics and offline trajectory planning. In Xu et al. [33], the authors formulate piano playing as an RL problem for a single Allegro hand on a miniature piano and leverage tactile sensor feedback. The piano playing tasks considered in these prior works are relatively simple (e.g., play up to six succes-

sive notes, or three successive chords with only two simultaneous keys pressed for each chord). On the other hand, ROBOPIANIST allows a general bi-manual controllable agent to emulate a pianist's growing proficiency by providing a large range of musical pieces with graded difficulties.

## 3  Experimental Setup

In this section, we introduce the simulated piano-playing environment as well as the musical suite used to train and evaluate our agent.

**Simulation details.**    We build our simulated piano-playing environment (depicted in Figure 1) using the open-source MuJoCo [34, 35] physics engine. The piano model is a full-size digital keyboard with 52 white keys and 36 black keys. We use a Kawai manual [36] as reference for the keys' positioning and dimensions. Each key is modeled as a joint with a linear spring and is considered "active" when its joint position is within $0.5°$ of its maximum range, at which point a synthesizer is used to generate a musical note. We also implement a mechanism to sustain the sound of any currently active note to mimic the mechanical effect of a sustain pedal on a real piano. The left and right hands are Shadow Dexterous Hand [37] models from MuJoCo Menagerie [38], which have been designed to closely reproduce the kinematics of the human hand.

**Musical representation.**    We use the Musical Instrument Digital Interface (MIDI) standard to represent a musical piece as a sequence of time-stamped messages corresponding to `note-on` or `note-off` events. A message carries additional pieces of information such as the pitch of a note and its velocity. We convert the MIDI file into a time-indexed note trajectory (a.k.a, piano roll), where each note is represented as a one-hot vector of length 88. This trajectory is used as the goal representation for the agent, informing it which keys must be pressed at each time step.

**Musical evaluation.**    We use precision, recall, and F1 scores to evaluate the proficiency of our agent. These metrics are computed by comparing the state of the piano keys at every time step with the corresponding ground-truth state, averaged across all time steps. If at any given time there are keys that should be "on" and keys that should be "off", precision measures how good the agent is at not hitting any of the "off" keys, while recall measures how good the agent is at hitting the "on" keys. The F1 score combines precision and recall into one metric, and ranges from 0 (if either precision or recall is 0) to 1 (perfect precision and recall). We primarily use the F1 score for our evaluations as it is a common heuristic accuracy score in the audio information retrieval literature [39], and we found empirically that it correlates with qualitative performance on our tasks.

**MDP formulation.**    We model piano-playing as a finite-horizon Markov Decision Process (MDP) defined by a tuple $(\mathcal{S}, \mathcal{A}, \rho, p, r, \gamma, H)$ where $\mathcal{S} \subset \mathbb{R}^n$ is the state space, $\mathcal{A} \subset \mathbb{R}^m$ is the action space, $\rho(\cdot)$ is the initial state distribution, $p(\cdot|s, a)$ governs the dynamics, $r : \mathcal{S} \times \mathcal{A} \to \mathbb{R}$ defines the rewards, $\gamma \in [0, 1)$ is the discount factor, and $H$ is the horizon. The goal of an agent is to maximize its total expected discounted reward over the horizon: $\mathbb{E}\left[\sum_{t=0}^{H} \gamma^t r(s_t, a_t)\right]$.

The agent's observations consist of proprioceptive and goal state information. The proprioceptive state contains hand and keyboard joint positions. The goal state information contains a vector of key goal states obtained by indexing the piano roll at the current time step, as well as a discrete vector indicating which fingers of the hands should be used at that timestep. To successfully play the piano, the agent must be aware of at least a few seconds' worth of its next goals in order to be able to plan appropriately. Thus the goal state is stacked for some lookahead horizon $L$. A detailed description of the observation space is given in Table 1. The agent's action is 45 dimensional and consists of target joint angles for the hand with an additional scalar value for the sustain pedal. The agent predicts

| Observations | Unit | Size |
|---|---|---|
| Hand and forearm joints | rad | 52 |
| Forearm Cartesian position | m | 6 |
| Piano key joints | rad | 88 |
| Active fingers | discrete | $L \cdot 10$ |
| Piano key goal state | discrete | $L \cdot 88$ |

Table 1: The agent's observation space. $L$ corresponds to the lookahead horizon.

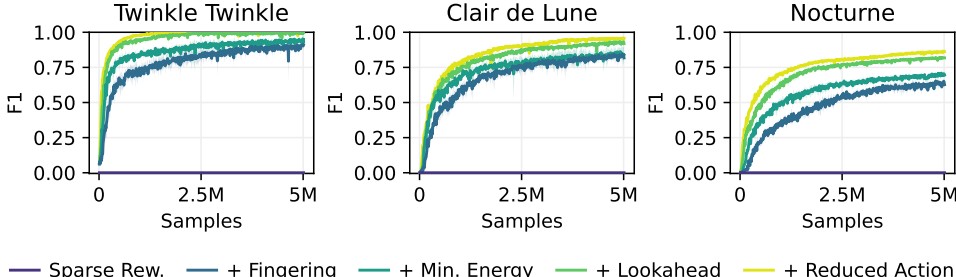

Figure 2: **ROBOPIANIST Design Considerations**. The F1 performance for 3 songs of increasing difficulty. From left to right as specified by the legend, each curve **inherits all MDP attributes of the curve before it** but makes one additional modification as described by its label.

target joint angles at 20Hz and the targets are converted to torques using PD controllers running at 500Hz. Since the reward function is crucial for learning performance, we discuss its design in Section 4.

**Fingering labels and dataset.** Piano fingering refers to the assignment of fingers to notes, e.g., "C4 played by the index finger of the right hand". Sheet music will typically provide sparse fingering labels for the tricky sections of a piece to help guide pianists, and pianists will often develop their own fingering preferences for a given piece. Since fingering labels aren't available in MIDI files by default, we used annotations from the PIG dataset [40] to create a corpus of 150 annotated MIDI files for use in the simulated environment. Overall this dataset we call REPERTOIRE-150 contains piano pieces from 24 Western composers spanning baroque, classical and romantic periods. The pieces vary in difficulty, ranging from relatively easy (e.g., Mozart's Piano Sonata K 545 in C major) to significantly harder (e.g., Scriabin's Piano Sonata No. 5) and range in length from tens of seconds to over 3 minutes.

## 4 ROBOPIANIST System Design

Our aim is to enable robots to exhibit sophisticated, high-dimensional control necessary for successfully performing challenging musical pieces. Mastery of the piano requires (i) spatial and temporal precision (hitting the right notes, at the right time), (ii) coordination (simultaneously achieving multiple different goals, in this case, fingers on each hand hitting different notes, without colliding), and (iii) planning (how a key is pressed should be conditioned on the expectation of how it would enable the policy to reach future notes). These behaviors do not emerge if we solely optimize with a sparse reward for pressing the right keys at the right times. The main challenge is exploration, which is further exacerbated by the high-dimensional nature of the control space.

We overcome this challenge with careful system design and human priors, which we detail in this section. The main results are illustrated in Figure 2. We pick 3 songs in increasing difficulty from ROBOPIANIST-REPERTOIRE-150. We note that "Twinkle Twinkle" is the easiest while "Nocturne" is the hardest. We train for 5M samples, 3 seeds. We evaluate the F1 every 10K training steps for 1 episode (no stochasticity in the environment).

### 4.1 Human priors

We found that the agent struggled to play the piano with a sparse reward signal due to the exploration challenge associated with the high-dimensional action space. To overcome this issue, we incorporated the fingering labels within the reward formulation (Table 2). When we remove this prior and only reward the agent for the key press, the agent's F1 stays at zero and no substantial learning progress is made. We suspect that the benefit of fingering comes not only from helping the agent achieve the current goal, but facilitating key presses in subsequent timesteps. Having the policy discover its own preferred fingering, like an experienced pianist, is an exciting direction for future research.

## 4.2 Reward design

We first include a reward proportional to how depressed the keys that should be active are. We then add a constant penalty if any inactive keys are pressed hard enough to produce sound. This gives the agent some leeway to rest its fingers on inactive keys so long as they don't produce sound. We found that giving a constant penalty regardless of the number of false positives was crucial for learning; otherwise, the agent would become too conservative and hover above the keyboard without pressing any keys. In contrast, the smooth reward for pressing active keys plays an important role in exploration by providing a dense learning signal. We introduce two additional shaping terms: (i) we encourage the fingers to be spatially close to the keys they need to press (as prescribed by the fingering labels) to help exploration, and (ii) we minimize energy expenditure, which reduces variance (across seeds) and erratic behavior control policies trained with RL are prone to generate. The total reward at a given time step is a weighted sum over the aforementioned components. A detailed description of the reward function can be found in Appendix B.

## 4.3 Peeking into the future

We observe additional improvements in performance and variance from including future goal states in the observation, i.e., increasing the lookahead horizon $L$. Intuitively, this allows the policy to better plan for future notes – for example by placing the non-finger joints (e.g., the wrist) in a manner that allows more timely reaching of notes at the next timestep.

## 4.4 Constraining the action space

To alleviate exploration even further, we explore disabling degrees of freedom [41] in the Shadow Hand that either do not exist in the human hand (e.g., the little finger being opposable) or are not strictly necessary for most songs. We additionally reduce the joint range of the thumb. While this speeds up learning considerably, we observe that with additional training time, the full action space eventually achieves similar F1 performance.

# 5 Results

In this section, we present our experimental findings on ROBOPIANIST-ETUDE-12, a subset of ROBOPIANIST-REPERTOIRE-150 consisting of 12 songs. The results on the full ROBOPIANIST-REPERTOIRE-150 can be found in Appendix C. We design our experiments to answer the following questions:

(1) How does our method compare to a strong baseline in being able to play individual pieces?
(2) How can we enable a single policy to learn to play more than one song?
(3) What effects do our design decisions have on the feasibility of acquiring highly complex, dexterous control policies?

## 5.1 Specialist Policy Learning

For our policy optimizer, we use a state-of-the-art model-free RL algorithm DroQ [42], one of several regularized variants of the widely-used Soft-Actor-Critic [43] algorithm. We evaluate online predictive control (MPC) as a meaningful point of comparison. Specifically, we use the implementation from Howell et al. [44] that leverages the physics engine as a dynamics model, and which was shown to solve previously-considered-challenging dexterous manipulation tasks [9, 13, 11] in simulation. Amongst various planner options in [44], we found the most success with `Predictive Sampling`, a derivative-free sampling-based method. A detailed discussion of this choice of baseline and its implementation can be found in Appendix F. Our method uses 5 million samples to train for each song using the same set of hyperparameters, and the MPC baseline is run at one-tenth of real-time speed to give the planner adequate search time.

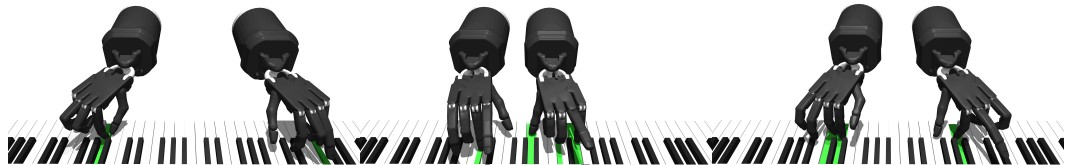

Figure 3: Policies in the ROBOPIANIST environment displaying skilled piano behaviors such as (left) simultaneously controlling both hands to reach for notes on opposite ends of the keyboard, (middle) playing a chord with the left hand by precisely and simultaneously hitting a note triplet, and (right) playing a trill with the right hand, which involves rapidly alternating between two adjacent notes.

The quantitative results are shown in Figure 4. We observe that the ROBOPIANIST agent significantly outperforms the MPC baseline, achieving an average F1 score of 0.79 compared to 0.43 for MPC. We hypothesize that the main bottleneck for MPC is compute: the planner struggles with the large search space which means the quality of the solutions that can be found in the limited time budget is poor. Qualitatively, our learned agent displays remarkably skilled piano behaviors such as (1) simultaneously controlling both hands to reach for notes on opposite ends of the keyboard, (2) playing *chords* by precisely and simultaneously hitting note triplets, and (3) playing *trills* by rapidly alternating between adjacent notes (see Figure 3). We encourage the reader to listen to these policies on the supplementary website[1].

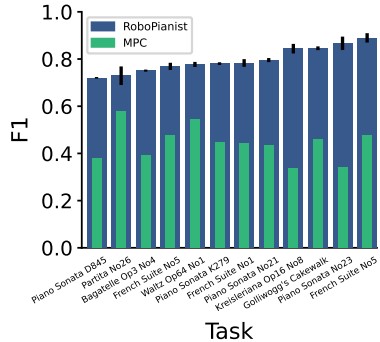

Figure 4: The F1 scores achieved by ROBOPIANIST (blue) and MPC (green) for each of the ROBOPIANIST-ETUDE-12 tasks.

## 5.2 Multi-Song Policy Learning

**Multitask RL**   Ideally, a single agent should be able to learn how to play all songs. Secondly, given enough songs to practice on, we would like this agent to zero-shot generalize to new ones. To investigate whether this is possible, we create a multi-task environment where the number of tasks in the environment corresponds to the number of songs available in the training set. Note that in this multi-song setting, environments of increasing size are additive (i.e., a 2-song environment contains the same song as the 1-song environment plus an additional one). We use Für Elise as the base song and report the performance of a single agent trained on increasing amounts of training songs in Figure 5. We observe that training on an increasing amount of songs is significantly harder than

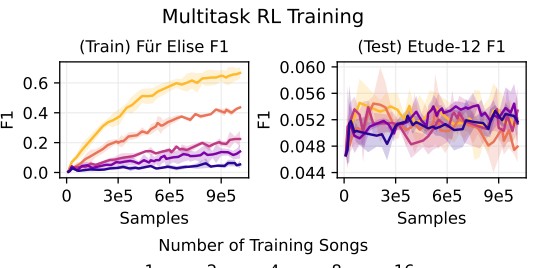

Figure 5: **Multi-song training with RL**. We test the ability to learn multiple songs simultaneously. The training performance on the shared song, Für Elise, is shown on the left. We see adding more songs degrades performance on individual songs. On the right, we evaluate on the held out ROBOPIANIST-ETUDE-12 songs, suggesting the difficulty of generalization with multitask RL training.

training specialist policies on individual songs. Indeed, the F1 score on Für Elise continues to drop as the number of training songs increases, from roughly 0.7 F1 for 1 song (i.e., the specialist) to almost 0 F1 for 16 songs. We also evaluate the agent's zero-shot performance (i.e., no fine-tuning) on ROBOPIANIST-ETUDE-12 (which does not have overlap with the training songs) to test our RL agent's ability to generalize. We see, perhaps surprisingly, that multitask RL training fails to positively transfer on the test set regardless of the size of the pre-training tasks.

---

[1] https://kzakka.com/robopianist/

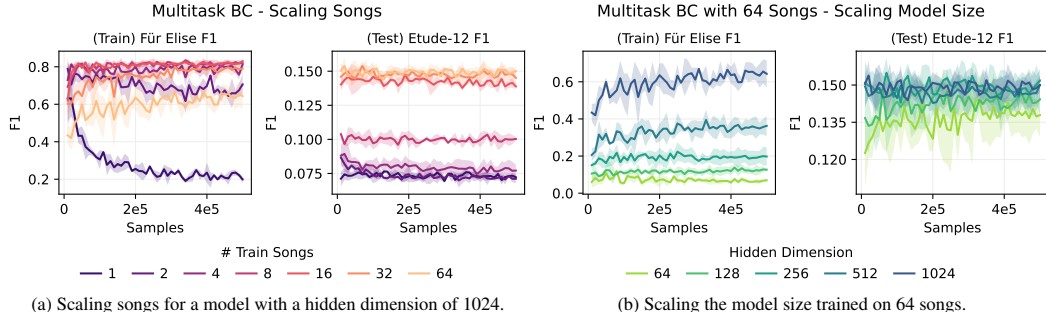

(a) Scaling songs for a model with a hidden dimension of 1024.  (b) Scaling the model size trained on 64 songs.

Figure 6: Here we show the learning curves of BC, showing the interplay between the attempted number of training songs and model size. The song Für Elise, is in the training dataset for each experiment. We show the F1 score across Für Elise and ROBOPIANIST-ETUDE-12. We see that for a small number of songs, the model quickly overfits, suggesting that for the given model size the amount of data and diversity is lacking. We see that adding more songs results in a higher test F1, suggesting that the model is beginning to generalize. However, zero-shot test performance is far from that of the expert.

As we increase the model size, the F1 score on the training song improves, suggesting that larger models can better capture the complexity of the piano playing task across multiple songs.

**Multitask Behavioral Cloning**   Since multitask training with RL is challenging, we instead distill the specialist RL policies trained in Subsection 5.1 into a single multitask policy with Behavioral Cloning (BC) [45].  To do so, we collect 100 trajectories for each song in the ROBOPIANIST-REPERTOIRE-150, hold out 2 for validation, and use the ROBOPIANIST-ETUDE-12 trajectories for testing.  We then train a feed-forward neural network to predict the expert actions conditioned on the state and goal using a mean squared error loss.  For a more direct comparison with multi-task RL, the dataset subsets use trajectories from the same songs used in the equivalently sized multi-task RL experiments. We observe in Figure 6a that as we increase the number of songs in the training dataset, the model's ability to generalize improves, resulting in higher test F1 scores. We note that for a large model (hidden size of 1024), training on too few songs results in overfitting because there isn't enough data. Using a smaller model alleviates this issue, as shown in the more detailed multitask BC results found in Appendix E, but smaller models are unable to perform well on multiple songs. Despite having better generalization performance than RL, zero-shot performance on ROBOPIANIST-ETUDE-12 falls far below the performance achieved by the specialist policy. Additionally, we investigate the effect of model size on the multitask BC performance. We train models with different hidden dimensions (fixing the number of hidden layers) on a dataset of 64 songs. As shown in Figure 6b, a smaller hidden dimension results in lower F1 performance which most likely indicates underfitting.

## 5.3   Further analysis

In this section, we discuss the effect of certain hyperparameters on the performance of the RL agent.

**Control frequency and lookahead horizon:**   Figure 7 illustrates the interplay between the control frequency (defined as the reciprocal of control timestep) and the lookahead horizon $L$, and their effect on the F1 score. Too large of a control timestep can make it impossible to play faster songs, while a smaller control timestep increases the effective task horizon, thereby increasing the computational complexity of the task.  Lookahead on the other hand controls how far into the future the agent can see goal states. We observe that as the control timestep decreases, the lookahead must be increased to maintain the agent ability to see and reason about future goal states. A control frequency of 20Hz (0.05 seconds) is a sweet spot, with notably 100Hz (0.01 seconds) drastically reducing the final score.  At 100Hz, the MDP becomes too long-horizon, which complicates exploration, and at 10Hz, the discretization of the MIDI file becomes too coarse, which negatively impacts the timing of the notes.

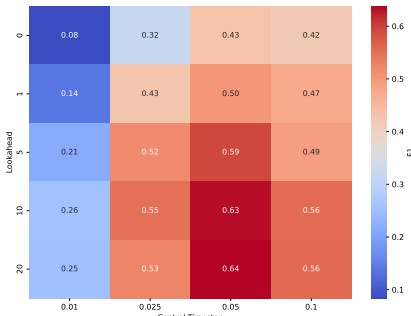

Figure 7: **Lookahead & Control Timestep**. These parameters jointly modulate the foresight and control granularity of the agent. We find a control timestep of 0.05 offers optimal control granularity, and performance generally improves with the lookahead up to a point.

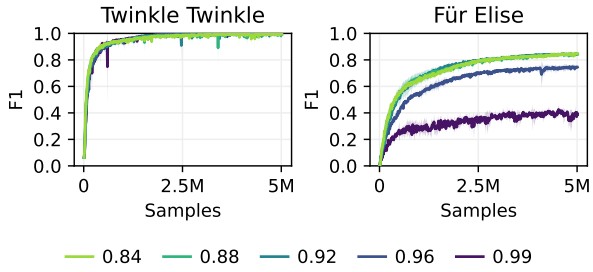

Figure 8: **Discount factor**. We find the discount factor to have a high impact on F1, especially as the task complexity increases. While on simpler songs (Twinkle Twinkle) the discount factor does not have any visible effect, it is critical for faster-paced songs (Für Elise).The curves show that any discount in the range of 0.84-0.92 can be usd with similar effect.

**Discount factor**: As shown in Figure 8, the discount factor has a significant effect on the F1 score. We sweep over a range of discount factors on two songs of varying difficulty. Notably, we find that discounts in the range $0.84$ to $0.92$ produce policies with roughly the same performance and high discount factors (e.g., 0.99, 0.96) result in lower F1 performance. Qualitatively, on Für Elise, we noticed that agents trained with higher discounts were often conservative, opting to skip entire note sub-segments. However, agents trained with lower discount factors were willing to risk making mistakes in the early stages of training and thus quickly learned how to correctly strike the notes and attain higher F1 scores.

## 6 Discussion

**Limitations**  While ROBOPIANIST produces agents that push the boundaries of bi-manual dexterous control, it does so in a simplified simulation of the real world. For example, the velocity of a note, which modulates the strength of the key press, is ignored in the current reward formulation. Thus, the dynamic markings of the composition are ignored. Furthermore, our RL training approach can be considered wasteful, in that we learn by attempting to play the entire piece at the start of every episode, rather than focusing on the parts of the song that need more practicing. Finally, our results highlight the challenges of multitask learning especially in the RL setting.

**Conclusion**  In this paper, we introduced ROBOPIANIST, which provides a simulation framework and suite of tasks in the form of a corpus of songs, together with a high-quality baseline and various axes of evaluation, for studying the challenging high-dimensional control problem of mastering piano-playing with two hands. Our results demonstrate the effectiveness of our approach in learning a broad repertoire of musical pieces, and highlight the importance of various design choices required for achieving this performance. There is an array of exciting future directions to explore with ROBOPIANIST including: leveraging human priors to accelerate learning (e.g., motion priors from YouTube), studying zero-shot generalization to new songs, incorporating multimodal data such as sound and touch. We believe that ROBOPIANIST serves as a valuable platform for the research community, enabling further advancements in high-dimensional control and dexterous manipulation.

**Acknowledgments**

We would like to thank members of the Robot Learning Lab and anonymous reviewers for their feedback and helpful comments. This project was supported in part by ONR #N00014-22-1-2121 under the Science of Autonomy program.

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

# A  Extended Related Work

We address related work within two primary areas: dexterous high-dimensional control, and robotic pianists.

**Dexterous Manipulation and High-Dimensional Control** The vast majority of the control literature uses much lower-dimensional systems (i.e., single-arm, simple end-effectors) than high-dimensional dexterous hands. Specifically, only a handful of general-purpose policy optimization methods have been shown to work on high-dimensional hands, even for a single hand [10, 9, 12, 11, 16, 14, 17, 7], and of these, only a subset has demonstrated results in the real world [10, 9, 12, 11, 16]. Results with bi-manual hands are even rarer, even in simulation only [15, 18].

As a benchmark, perhaps the most distinguishing aspect of ROBOPIANIST is in the definition of "task success". As an example, general manipulation tasks are commonly framed as the continual application of force/torque on an object for the purpose of a desired change in state (e.g., SE(3) pose and velocity). Gradations of dexterity are predominantly centered around the kinematic redundancy of the arm or the complexity of the end-effector, ranging from parallel jaw-grippers to anthropomorphic hands [46, 15]. A gamut of methods have been developed to accomplish such tasks, ranging from various combinations of model-based and model-free RL, imitation learning, hierarchical control, etc. [47, 10, 13, 12, 48, 49]. However, the class of problems generally tackled corresponds to a definition of dexterity pertaining to traditional manipulation skills [19], such as re-orientation, relocation, manipulating simply-articulated objects (e.g., door opening, ball throwing and catching), and using simple tools (e.g., hammer) [20, 21, 15, 11, 22]. The only other task suite that we know of that presents bi-manual tasks, the recent Bi-Dex [15] suite, presents a broad collection of tasks that fall under this category.

While these works represent an important class of problems, we explore an alternative notion of dexterity and success. In particular, for most all the aforementioned suite of manipulation tasks, the "goal" state is some explicit, specific geometric function of the final states; for instance, an open/closed door, object re-oriented, nail hammered, etc. This effectively reduces the search space for controls to predominantly a single "basin-of-attraction" in behavior space per task. In contrast, the ROBOPIANIST suite of tasks encompasses a more complex notion of a goal, which is encoded through a musical performance. In effect, this becomes a highly combinatorially variable sequence of goal states, extendable to arbitrary difficulty by only varying the musical score. "Success" is graded on accuracy over an entire episode; concretely, via a time-varying non-analytic output of the environment, i.e., the music. Thus, it is not a matter of the "final-state" that needs to satisfy certain termination/goal conditions, a criterion which is generally permissive of less robust execution through the rest of the episode, but rather the behavior of the policy *throughout the episode* needs to be precise and musical.

Similarly, the literature on humanoid locomotion and more broadly, "character control", another important area of high-dimensional control, primarily features tasks involving the discovery of stable walking/running gaits [50, 51, 52], or the distillation of a finite set of whole-body movement priors [53, 54, 55], to use downstream for training a task-level policy. Task success is typically encoded via rewards for motion progress and/or reaching a terminal goal condition. It is well-documented that the endless pursuit of optimizing for these rewards can yield unrealistic yet "high-reward" behaviors. While works such as [53, 56] attempt to capture *stylistic* objectives via leveraging demonstration data, these reward functions are simply appended to the primary task objective. This scalarization of multiple objectives yields an arbitrarily subjective Pareto curve of optimal policies. In contrast, performing a piece of music entails both objectively measurable precision with regards to melodic and rhythmic accuracy, as well as a subjective measure of musicality. Mathematically, this translates as *stylistic* constraint satisfaction, paving the way for innovative algorithmic advances.

**Robotic Piano Playing** Robotic pianists have a rich history within the literature, with several works dedicated to the design of specialized hardware [23, 24, 25, 26, 27, 28], and/or customized controllers for playing back a song using pre-programmed commands (open-loop) [29, 30]. The work

in [31] leverages a combination of inverse kinematics and trajectory stitching to play single keys and playback simple patterns and a song with a Shadow hand [37]. More recently, in [32], the author simulated robotic piano playing using offline motion planning with inverse kinematics for a 7-DoF robotic arm, along with an Iterative Closest Point-based heuristic for selecting fingering for a four-fingered Allegro hand. Each hand is simulated separately, and the audio results are combined post-hoc. Finally, in [33], the authors formulate piano playing as an RL problem for a single Allegro hand (four fingers) on a miniature piano, and additionally leverage tactile sensor feedback. However, the tasks considered are rather simplistic (e.g., play up to six successive notes, or three successive chords with only two simultaneous keys pressed for each chord). The ROBOPIANIST benchmark suite is designed to allow a general bi-manual controllable agent to emulate a pianist's growing proficiency on the instrument by providing a curriculum of musical pieces, graded in difficulty. Leveraging two underactuated anthropomorphic hands as actuators provides a level of realism and exposes the challenge of mastering this suite of high-dimensional control problems.

# B    Detailed Reward Function

| Reward | Formula | Weight | Explanation |
|---|---|---|---|
| Key Press | $0.5 \cdot g(||k_s - k_g||_2) + 0.5 \cdot (1 - \mathbf{1}_{\{\text{false positive}\}})$ | 1 | Press the right keys and only the right keys |
| Energy Penalty | $|\tau_{\text{joints}}|^\top |v_{\text{joints}}|$ | -5e-3 | Minimize energy expenditure |
| Finger Close to Key | $g(||p_f - p_k||_2)$ | 1 | Shaped reward to bring fingers to key |

Table 2: The reward function used to train ROBOPIANIST agents. $\tau$ represents the joint torque, $v$ is the joint velocity, $p_f$ and $p_k$ represent the position of the finger and key in the world frame respectively, $k_s$ and $k_g$ represent the current and the goal states of the key respectively, and $g$ is a function that transforms the distances to rewards in the $[0, 1]$ range.

## C Full Repertoire Results

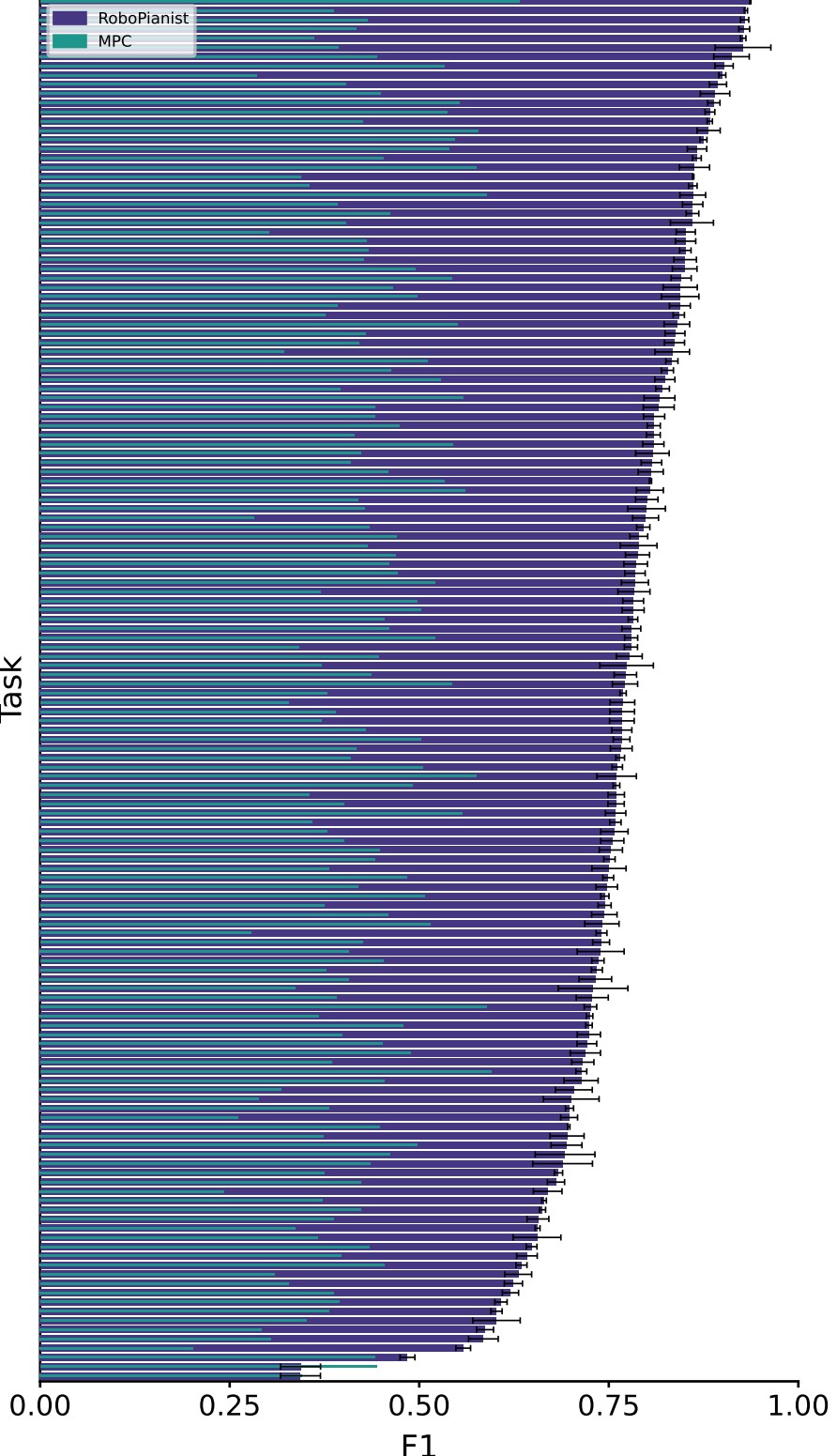

Figure 9: Results on the full repertoire of 150 songs.

# D ROBOPIANIST Training details

**Computing infrastructure and experiment running time**

Our model-free RL codebase is implemented in JAX [57]. Experiments were performed on a Google Cloud `n1-highmem-64` machine with an Intel Xeon E5-2696V3 Processor hardware with 32 cores (2.3 GHz base clock), 416 GB RAM and 4 Tesla K80 GPUs. Each "run", i.e., the training and evaluation of a policy on one task with one seed, took an average of 5 hrs wall clock time. These run times are recorded while performing up to 8 runs in parallel.

**Network architecture**

We use a regularized variant of clipped double Q-learning [58, 59], specifically DroQ [42], for the critic. Each $Q$-function is parameterized by a 3-layer multi-layer perceptron (MLP) with `ReLU` activations. Each linear layer is followed by dropout [60] with a rate of $0.01$ and layer normalization [61]. The actor is implemented as a `tanh`-diagonal-Gaussian, and is also parameterized by a 3-layer MLP that outputs a mean and covariance. Both actor and critic MLPs have hidden layers with 256 neurons and their weights are initialized with Xavier initialization [62], while their biases are initialized to zero.

**Training and evaluation**

We first collect 5000 seed observations with a uniform random policy, after which we sample actions using the RL policy. We then perform one gradient update every time we receive a new environment observation. We use the Adam [63] optimizer for neural network optimization. Evaluation happens in parallel in a background thread every 10000 steps. The latest policy checkpoint is rolled out by taking the mean of the output (i.e., no sampling). Since our environment is "fixed", we perform only one rollout per evaluation.

**Reward formulation**

The reward function for training the RL agent consists of three terms: 1) a key press term $r_{\text{key}}$, 2) a move finger to key term $r_{\text{finger}}$, and 3) an energy penalty term $r_{\text{energy}}$.

$r_{\text{key}}$ encourages the policy to press the keys that need to be pressed and discourages it from pressing keys that shouldn't be pressed. It is implemented as:

$$r_{\text{key}} = 0.5 \cdot \left( \frac{1}{K} \sum_{i}^{K} g(||k_s^i - 1||_2) \right) + 0.5 \cdot (1 - \mathbf{1}_{\{\text{false positive}\}}),$$

where K is the number of keys that need to be pressed at the current timestep, $k_s$ is the normalized joint position of the key between 0 and 1, and $\mathbf{1}_{\{\text{false positive}\}}$ is an indicator function that is 1 if any key that should not be pressed creates a sound. $g$ is the `tolerance` function from the `dm_control` [52] library: it takes the L2 distance of $k_s$ and 1 and converts it into a bounded positive number between 0 and 1. We use the parameters `bounds=0.05` and `margin=0.5`.

$r_{\text{finger}}$ encourages the fingers that are active at the current timestep to move as close as possible to the keys they need to press. It is implemented as:

$$r_{\text{finger}} = \frac{1}{K} \sum_{i}^{K} g(||p_f^i - p_k^i||_2),$$

where $p_f$ is the Cartesian position of the finger and $p_i$ is the Cartesian position of a point centered at the surface of the key. $g$ for this reward is parameterized by `bounds=0.01` and `margin=0.1`.

Finally, $r_{\text{energy}}$ penalizes high energy expenditure and is implemented as:

$$r_{\text{energy}} = |\tau_{\text{joints}}|^\top |\mathsf{v}_{\text{joints}}|,$$

where $\tau_{\text{joints}}$ is a vector of joint torques and $\mathsf{v}_{\text{joints}}$ is a vector of joint velocities.

The final reward function sums up the aforementioned terms as follows:

$$r_{\text{total}} = r_{\text{key}} + r_{\text{finger}} - 0.005 \cdot r_{\text{energy}}$$

**Other hyperparameters**

For a comprehensive list of hyperparameters used for training the model-free RL policy, see Table 3.

| Hyperparameter | Value |
|---|---|
| Total train steps | 5M |
| Optimizer | |
|    Type | ADAM |
|    Learning rate | $3 \times 10^{-4}$ |
|    $\beta_1$ | 0.9 |
|    $\beta_2$ | 0.999 |
| Critic | |
|    Hidden units | 256 |
|    Hidden layers | 3 |
|    Non-linearity | ReLU |
|    Dropout rate | 0.01 |
| Actor | |
|    Hidden units | 256 |
|    Hidden layers | 3 |
|    Non-linearity | ReLU |
| Misc. | |
|    Discount factor | 0.99 |
|    Minibatch size | 256 |
|    Replay period every | 1 step |
|    Eval period every | 10000 step |
|    Number of eval episodes | 1 |
|    Replay buffer capacity | 1M |
|    Seed steps | 5000 |
|    Critic target update frequency | 1 |
|    Actor update frequency | 1 |
|    Critic target EMA momentum ($\tau_Q$) | 0.005 |
|    Actor log std dev. bounds | $[-20, 2]$ |
|    Entropy temperature | 1.0 |
|    Learnable temperature | True |

Table 3: Hyperparameters for all model-free RL experiments.

# E   Multitask BC Results

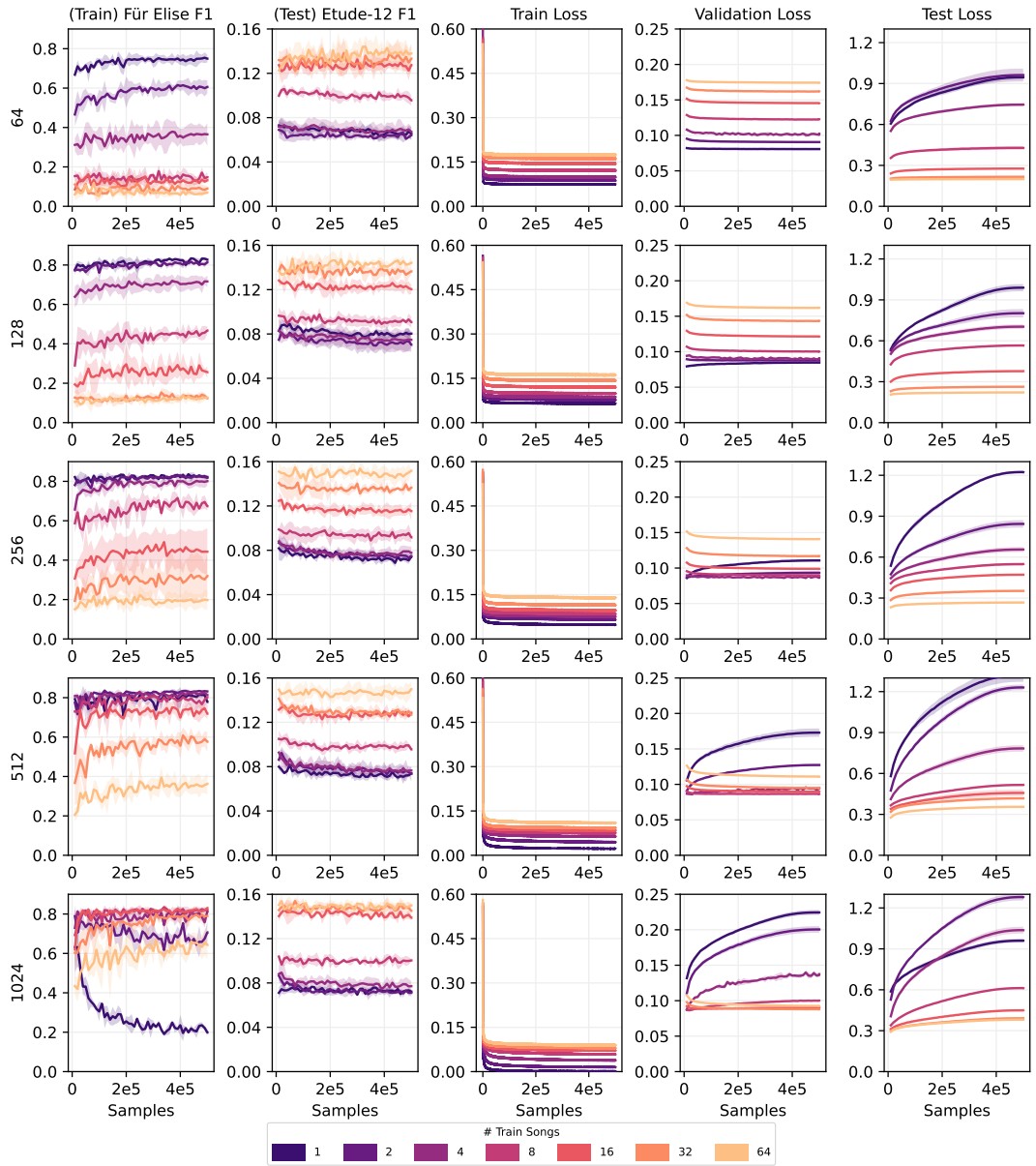

Figure 10: Caption for the figure.

# F    Baselines

**Computing infrastructure and experiment running time**

Our MPC codebase is implemented in C++ with MJPC [44]. Experiments were performed on a 2021 M1 Max Macbook Pro with 64 GB of RAM.

**Algorithm**

We use MPC with Predictive Sampling (PS) as the planner. PS is a derivative-free sampling-based algorithm that iteratively improves a nominal sequence of actions using random search. Concretely, $N$ candidates are created at every iteration by sampling from a Gaussian with the nominal as the mean and a fixed standard deviation $\sigma$. The returns from the candidates are evaluated, after which the highest scoring candidate is set as the new nominal. The action sequences are represented with cubic splines to reduce the search space and smooth the trajectory. In our experiments, we used

$N = 10$, $\sigma = 0.05$, and a spline dimension of 2. We plan over a horizon of 0.2 seconds, use a planning time step of 0.01 seconds and a physics time step of 0.005 seconds.

**Cost formulation**

The cost function for the MPC baseline consists of 2 terms: 1) a key press term $c_{\text{key}}$, 2) and a move finger to key term $c_{\text{finger}}$.

The costs are implemented similarly to the model-free baseline, but don't make use of the $g$ function, i.e., they solely consist in unbounded l2 distances.

The total cost is thus:

$$c_{\text{total}} = c_{\text{key}} + c_{\text{finger}}$$

Note that we experimented with a control cost and an energy cost but they decreased performance so we disabled them.

**Alternative baselines**

We also tried the optimized *derivative-based* implementation of iLQG [64] also provided by [44], but this was not able to make substantial progress even at significantly slower than real-time speeds. iLQG is difficult to make real time because the action dimension is large and the algorithm theoretical complexity is $O(|A|^3 \cdot H)$. The piano task presents additional challenges due to the large number of contacts that are generated at every time step. This make computing derivatives for iLQG very expensive (particularly for our implementation which used finite-differencing to compute them). A possible solution would be to use analytical derivatives and differentiable collision detection.

Besides online MPC, we could have used offline trajectory optimization to compute short reference trajectories for each finger press offline and then track these references online (in real time) using LQR. We note, however, that the (i) high dimensionality, (ii) complex sequence of goals adding many constraints, and (iii) overall temporal length (tens of seconds) of the trajectories pose challenges for this sort of approach.

