# OpenReview forum: "RoboPianist: Dexterous Piano Playing with Deep Reinforcement Learning"
_robot-learning.org/CoRL/2023/Conference — CoRL 2023 Poster_

### Official Review · Reviewer_F4CG · 2023-07-17

**Confidence:** 4
**Originality:** Fair
**Technical Quality:** Very Good
**Clarity Of Presentation:** Very Good
**Impact:** 3

**Recommendation:**

Weak Accept: I recommend accepting the paper, but will not argue for my recommendation if the majority of other reviewers have a different opinion.

**Review:**

### Strengths
- The paper is looking at a potentially profoundly interesting problem (i.e., robot pianist), which can go beyond just the topic of dexterous control but more deeply into the space of AI creativity and music performing arts. For example, can a robot play classical pieces in idiosyncratic but acoustically pleasant manner? Progress on this front could even make impact in the music domain ultimately.
- Experiments are thorough and well executed. Technical and experimental details are well presented.
- The paper is quite open and clear about the limitations of it's approach and the limitations of the results.
- Overall, the paper is well written and a pleasant read. Figures and illustrations are beautifully made.

### Weaknesses
- Certain contributions are over-claimed and not well-supported with proper empirical evidence. The abstract claims that the proposed method can "produce visually and acoustically pleasing performances" (L11). The same claim is also made in the intro section (L42: "... producing visually and acoustically pleasing performances,"). But there is no evaluation nor any reported metrics on visual or acoustical "pleasantness". It is also rather counter-intuitive that DRL can learn "pleasing" performance without training with any human feedback.
- Following the last point, one expected metric for "naturalness" or "pleasantness" is through human evaluation. However, no human evaluation has been provided. Also, from my little personal biased view, I find the provided examples, though impressive in some way, still not quite "acoustically pleasant" yet. It still makes some entry-level mistakes that are common among piano beginners.
- Without evaluating at the level of "artisticness" or "pleasantness", the task is only aimed to be solved at a "control" level, i.e. aligning key presses with the piano roll, which really downplays the artistic side of the task, and rather unuseful. One can build mechanical systems that can "play piano" with precise key strokes---why bother with human-like hands?
- Speaking of merely "pushing robot dexterity", the result is rather unexciting. Standard RL can nail the task to a large extent but that can also somewhat be foreseen. Generalization to unseen pieces would be a significant deal, but this paper is not there yet still.
- Also on the note of "pushing dexterity", the techniques presented in this paper seems very specific to the task of piano playing (e.g., state space and reward defined on key presses, fingering, etc.). It is unclear how easy to adapt when one wants to solve a different dexterous tasks.
- One little thought on the training of multitask RL: rather than "learning multiple songs simultaneously" (Fig. 5), why not "progressively". I.e., start from one song; once converged, add a new song and continue training on both songs, etc. I thought this was what the authors intended to do but it looks like Fig. 5 is comparing training in different $N$-song envs "from scratch".

### Question and Suggestions
- [L90] "We use the Musical Instrument Digital Interface (MIDI) standard to represent musical pieces and synthesize sounds." -> Sentence duplicated with L84-85.
- [L112] "We define actions to be target joint angles for the 22 joints in each hand ... convert them to torques at the 52 hand joints." -> $52 - 22 \times 2 = 8$. What do these 8 extra joints account for? Movement of the forearms' base?
- [Fig. 2] The term "Action Reward" is a bit confusing. Why not just use "Energy Penalization"?
- [L150] "... sheet music will typically provide sparse fingering information, ..." -> Is this what is actually used for "human-prior" fingering information? This can be made more explicit in the paragraph.
- [L266] "can result make it impossible" -> "can make it impossible"
- [Fig. 7] What is the unit of y-axis (Lookahead)? Timesteps (i.e., 20 means 1 second for 20 Hz control timestep, which seems a bit short for planning)? Or seconds (whereas 20 seconds seems a bit long)?

**Quality Of The Limitations Section:**

Limitations are addressed clearly

**Questions For Rebuttal:**

Address the weaknesses listed above.

**Robotics Focus:**

Highly relevant to robotics but no hardware experiments

**Summary Of Paper:**

- This paper aims to challenge learning-based dexterity with the task of piano playing with bi-manual anthropomorphic hands in simulation.
- A new simulation environment is created, powered by the MuJoCo engine with two Shadow Hands and a simulated piano.
- A state space and reward function for RL is designed and justified.
- Evaluation is based on the alignment between key presses and the "piano rolls" extracted from the corresponding MIDI files.
- A dataset of 150 music pieces is constructed for training and evaluation.
- A standard RL algorithm (DroQ) is adopted for policies training.
- Different training paradigms are explored, including training one policy per piece, training one policy for multiple pieces, and training per piece experts followed by distillation to multi-piece policies using behavior cloning (BC).
- Multitask RL polices struggles with training, while per task policies and multitask BC can excel to a large extent.
- Zero-shot generalization remains challenging.

**Summary Of Recommendation:**

The presentation and experiments look very well executed. However, I struggle see a strong merit otherwise. The task itself, without human evaluation, is questionable. Methodology-wise the contribution is also limited and specific to the piano playing problem.

### Post Rebuttal
The rebuttal addresses my concern on the over-claimed contributions. While falling short on some of the interesting aspects (e.g. "aligning with human values of pleasantness" and "generalizing to new songs") at its current form, I agree with the authors that the publication of this work can lead to future development in these areas. I also like the "unique dimension to policy evaluation" ("does my policy sound good") of the piano task. Therefore I'm willing to raise my recommendation from weak reject to weak accept.

---

> ### Author Response · Authors · 2023-08-06
> **Response to reviewer F4CG (1/2)**
>
> Thank you for taking the time to review our paper and for providing helpful comments! We are happy to hear that you found the problem setting interesting, the experiments thorough and well-executed and the paper well-written and pleasant to read. You brought up some critical questions which we discuss below. Please let us know if you have any remaining questions or concerns.
>
> > Certain contributions are over-claimed and not well-supported with proper empirical evidence. The abstract claims that the proposed method can "produce visually and acoustically pleasing performances" (L11). The same claim is also made in the intro section (L42: "... producing visually and acoustically pleasing performances,"). But there is no evaluation nor any reported metrics on visual or acoustical "pleasantness". It is also rather counter-intuitive that DRL can learn "pleasing" performance without training with any human feedback. Following the last point, one expected metric for "naturalness" or "pleasantness" is through human evaluation. However, no human evaluation has been provided. Also, from my little personal biased view, I find the provided examples, though impressive in some way, still not quite "acoustically pleasant" yet. It still makes some entry-level mistakes that are common among piano beginners.
>
> We agree that the term "acoustically pleasing" is subjective. We used this term with the intent to describe the fact that the agent can produce music that follows the intended rhythm and melody of the original piece, as indicated by the F1 score. However, we understand that "pleasing" can imply a level of artistic interpretation that our agent does not currently possess. As such, we will remove such statements from the final version of the paper. That being said, we think the RoboPianist task offers a unique dimension to policy evaluation (“does my policy sound good?”), and this can be done independently of the particular reward function used.
>
> > Without evaluating at the level of "artisticness" or "pleasantness", the task is only aimed to be solved at a "control" level, i.e. aligning key presses with the piano roll, which really downplays the artistic side of the task, and rather unuseful. One can build mechanical systems that can "play piano" with precise key strokes---why bother with human-like hands? Speaking of merely "pushing robot dexterity", the result is rather unexciting. Standard RL can nail the task to a large extent but that can also somewhat be foreseen. Generalization to unseen pieces would be a significant deal, but this paper is not there yet still.
>
> One of the goals of this work is to push the boundaries of what is possible with dexterous manipulation in a complex, high-dimensional task. While a mechanical system could indeed play the piano with precise keystrokes, controlling anthropomorphic hands to perform the same task is challenging. This work is not just about playing the piano but also demonstrating the potential for robots to perform tasks that require a high level of dexterity and precision.
>
> A significant part of the contribution of our work is the development of the RoboPianist benchmark which contains the ROBOPIANIST-REPERTOIRE-150 set of songs. While we only study the problem of learning a policy to play the song at a  “control” level, we believe that the benchmark itself provides more opportunities for different challenging research directions, such as aligning with human values of pleasantness or generalizing to new songs. We’re excited to see follow-up works address this!
>
> > Also on the note of "pushing dexterity", the techniques presented in this paper seems very specific to the task of piano playing (e.g., state space and reward defined on key presses, fingering, etc.). It is unclear how easy to adapt when one wants to solve a different dexterous tasks.
>
> While the specific reward function and state space we used are indeed tailored to the task of piano playing, the underlying approach of using deep RL to train an agent to perform a complex task is broadly applicable. The lessons we learned about the importance of careful reward shaping and the challenges of multitask learning can be applied to other tasks as well. We acknowledge that these findings are not unique but these are also not the main contributions of the paper. We believe that this work on piano playing can add to the body of research in learning dexterous policies as a valuable case study.

---

> > ### Author Response · Authors · 2023-08-06
> > **Response to reviewer F4CG (2/2)**
> >
> > > One little thought on the training of multitask RL: rather than "learning multiple songs simultaneously" (Fig. 5), why not "progressively". I.e., start from one song; once converged, add a new song and continue training on both songs, etc. I thought this was what the authors intended to do but it looks like Fig. 5 is comparing training in different
> > -song envs "from scratch".
> >
> > This is an excellent suggestion and one that we explored. Our finding was that if we trained on song B after the agent had learned song A, its performance at convergence on song B was worse than that of a policy trained on song B from scratch. As mentioned in the limitations, we think there is a lot of room for improvement in designing a curriculum to enable effective continual learning of songs.
> >
> > > What do these 8 extra joints account for? Movement of the forearms' base?
> >
> > The distinction here is joints vs actuators. The Shadow Hand has 24 joints but it is underactuated, so there are only 20 motors. Then there’s an additional 2 degrees of freedom for the translational movements of the forearm. Thus 22x2=44 actions that get converted to 52=((24+2) * 2) torques at all joints.
> >
> > > [Fig. 2] The term "Action Reward" is a bit confusing. Why not just use "Energy Penalization"?
> >
> > The “action reward” curve refers to an experiment where we append the action and reward obtained at the previous time step to the observation. We have removed it from Figure 2 and put it in the appendix. Please see the website ([robopianist.github.io](https://robopianist.github.io/)) for the updated figure.
> >
> > > [L150] "... sheet music will typically provide sparse fingering information, ..." -> Is this what is actually used for "human-prior" fingering information? This can be made more explicit in the paragraph.
> >
> > We use the same fingering information, but there are labels for every note as provided by the PIG dataset in our case.
> >
> > > [Fig. 7] What is the unit of y-axis (Lookahead)? Timesteps (i.e., 20 means 1 second for 20 Hz control timestep, which seems a bit short for planning)? Or seconds (whereas 20 seconds seems a bit long)?
> >
> > The lookahead is defined in units of steps and gets converted to seconds based on the control frequency. For example, a lookahead value of 20 with a control frequency of 20Hz (i.e., 0.05 s) translates to a lookahead of 1 second.
> >
> > > [L266] "can result make it impossible" -> "can make it impossible". [L90] "We use the Musical Instrument Digital Interface (MIDI) standard to represent musical pieces and synthesize sounds." -> Sentence duplicated with L84-85.
> >
> > We’ll fix these typos for the final version of the paper.
> >
> > Thanks again for your time reviewing our paper and for the insightful question. We hope that we have addressed your concerns; please let us know if you have any remaining questions.

---

> > > ### Author Response · Authors · 2023-08-09
> > > **Re: Rebuttal by authors**
> > >
> > > We again thank the reviewer for the constructive feedback. Please let us know if our response has addressed your concerns about the paper; we are happy to clarify any other concerns/questions that may be remaining.

---

> > > > ### Comment · Reviewer_F4CG · 2023-08-16
> > > > **Response to rebuttal**
> > > >
> > > > Thank you for your rebuttal response!

---

> ### Author Response · Authors · 2023-08-11
> **Friendly ping**
>
> Just a friendly ping in case this slipped through the cracks. We'd really appreciate a response so we can improve our submission. We appreciate your time and constructive feedback.

---

### Official Review · Reviewer_28kq · 2023-07-19

**Confidence:** 4
**Originality:** Good
**Technical Quality:** Good
**Clarity Of Presentation:** Very Good
**Impact:** 3

**Recommendation:**

Weak Accept: I recommend accepting the paper, but will not argue for my recommendation if the majority of other reviewers have a different opinion.

**Review:**

#### Strengths:

- The paper is well-written and the introduction section illustrates clarity about why this problem is important and challenging. Moreover, the problem setting is interesting and different, compared to a majority of tasks usually tackled under the umbrella of dexterous manipulation.
The simulation is built and released with an open-source physics simulator (MuJoCo), that makes it easier to reproduce and build on the proposed approach.
- The Robopianist-repertoire-150 is open-sourced, which helps in evaluating the proposed results in detail and also provides a good benchmark for future research in this direction.
- The paper discusses the generalization capabilities of the proposed framework in detail. While they show that multi-task RL does not show any benefit for generalization, multi-task behavioral cloning policy performs well and does show generalization capabilities with more data and larger models.
- The example result provided is an impressive qualitative depiction of the proposed methodology.

#### Weaknesses:

- The results on the website, although impressive, only show very few examples in a single video. Moreover,  the link provided does not provide code or dataset, unliked claimed in the abstract.
- The demo video does not mention the playback speed, which makes it harder to infer the actual efficacy and latency of the system. Also, no failure results are discussed on published on the website. It would be good if the author can provide a more detailed analysis of these results, commenting on when and why the system fails to better understand the current shortcomings.
- The current system assumes the availability of the fingering pattern for finger-to-key assignment, similar to (speed) typing. As pointed out by the author, without this information it is hard to optimize for emergent behaviors with sparse rewards. Although this can result in limited behavior emergence through deep RL and is left for future work to explore, it would be good to see some baseline results as an ablation study without these assumptions. This can provide a better picture for qualitative and quantitative comparisons, and also justify the proposed design choices more profoundly.
- The MPC baseline in Figure 3 is hypothesized to be much worse due to the computational bottleneck of the planner. Can the author explain and show if MPC works comparably or not with the RoboPianist for easier song pieces with fewer notes?
- No discussion is available for transfer to real world and the challenges that might accompany.

##### Minor errors:
- Figure 10 in the supplementary is missing a caption.
- L266 contains an error/type in the sentence - “A control time step that is too large can result in making it impossible…”


**Quality Of The Limitations Section:**

Limitations are addressed clearly

**Questions For Rebuttal:**

All the questions/comments that need to be addressed by the author are highlighted in the "Weaknesses" section.



**Robotics Focus:**

Highly relevant to robotics but no hardware experiments

**Summary Of Paper:**

The paper introduces a simulation setup for bi-manual anthropomorphic hands for the task of playing a piano. The task is intricate in itself, requiring coordination among the fingers, planning for efficiently performing the next set of actions, as well as spatial and temporal precision for hitting the right set of notes with appropriate intensity, simultaneously and at multiple stages. The key challenge as highlighted in the paper is the high-dimensional controls for dexterous manipulation which involve continuous on-and-off contact points between the piano keys and the robot hands. The paper proposed a deep reinforcement learning framework which can learn a collection of 150 piano pieces, called the ROBOPIANIST-REPERTOIRE-150 with both, single as well as multi-task policies which can learn to play more than 1 song in a single policy.

**Summary Of Recommendation:**

The paper poses an interesting problem setup, despite less novelty. It would be good to address the comments posted above. I recommend accepting the paper, but am keen to see some results with the real robot hands and piano setup.

---

> ### Author Response · Authors · 2023-08-06
> **Response to reviewer 28kq**
>
> Thank you for taking the time to review our paper and for providing helpful comments!  We are glad you found the paper well written, that RoboPianist is a good benchmark for future research and that the demo was an impressive qualitative depiction of the methodology.
>
> > The results on the website, although impressive, only show very few examples in a single video. Moreover, the link provided does not provide code or dataset, unliked claimed in the abstract.
>
> We’ve now updated the website ([https://robopianist.github.io](https://robopianist.github.io)) with links to the code and dataset, as well as videos for the full etude-12 subset (and more!).
>
> > The demo video does not mention the playback speed, which makes it harder to infer the actual efficacy and latency of the system.
>
> The policies are trained and executed at 20 Hz (i.e., the policy sends actions to the robot every 0.05 seconds). All videos on the website are played at real time (1x speed).
>
> > Also, no failure results are discussed on published on the website. It would be good if
>  the author can provide a more detailed analysis of these results, commenting on when and why the system fails to better understand the current shortcomings.
>
> Thank you for pointing this out. We now have explicit section on the website for common failure modes. Furthermore, we have an additional section titled “What makes a song hard” that computes various musical heuristics (e.g., notes per second, maximum polyphony, etc.) and looks at their correlation with the agent’s performance.
>
> > It would be good to see some baseline results as an ablation study without these assumptions.
>
> We completely agree and should have included the baseline in the submission. We’ve updated Figure 2 of the paper to include this baseline and it is now featured on the website.
>
> > No discussion is available for transfer to real world and the challenges that might accompany.
>
> We’ve expanded on this with a dedicated section on the website.
>
> We will include all updates and clarifications in the final version of the paper. Thanks again for your time reviewing our paper and for the helpful suggestions. We hope that we have addressed your concerns; please let us know if you have any remaining questions.

---

> > ### Author Response · Authors · 2023-08-09
> > **Re: Rebuttal by authors**
> >
> > We again thank the reviewer for the constructive feedback. Please let us know if our response has addressed your concerns about the paper; we are happy to clarify any other concerns/questions that may be remaining.

---

> > ### Author Response · Authors · 2023-08-11
> > **Friendly ping**
> >
> > Just a friendly ping in case this slipped through the cracks. We'd really appreciate a response so we can improve our submission. We appreciate your time and constructive feedback.

---

### Official Review · Reviewer_eFF7 · 2023-07-19

**Confidence:** 4
**Originality:** Very Good
**Technical Quality:** Very Good
**Clarity Of Presentation:** Excellent
**Impact:** 3

**Recommendation:**

Weak Accept: I recommend accepting the paper, but will not argue for my recommendation if the majority of other reviewers have a different opinion.

**Review:**

Strengths:
- This paper is dense and its contributions to the environment design for a novel task are significant. Designing the environment and task specification from scratch for such a complex task is a large effort. Its release will allow the community to study bimanual control with a dramatically lowered barrier to entry.
- The evaluation and ablations of the policy optimization and baseline are very thorough. They evaluate the impact of many interesting hyperparameters and design choices in good detail.
- The question of generalization across piano songs is interesting to explore. I agree with the assertion in Section 2 that traditional manipulation skills have a dominant "basin of attraction" while piano playing requires a more open-ended and diverse set of motor skills. And because the tasks are shown to be very diverse in difficulty and character, any method that improves generalization in this space must be more powerful. That seems like a quality of a good benchmark.

Weaknesses:
- The policy optimization approach departs from all prior references on learning dexterous manipulation by using DroQ instead of PPO. The motivation behind this design choice should be elaborated in the text.
- There is no sim-to-real evaluation. Also, no dynamics randomization, sensor noise, or lag modeling is mentioned. Also, the observation includes the piano key joints, which would be hard to sense in the real world. The benchmark could be strengthened by introducing these aspects. Otherwise, I am not sure if advancing progress on this benchmark brings us closer to a real robot that can play the piano.

**Quality Of The Limitations Section:**

Limitations are addressed clearly

**Questions For Rebuttal:**

- tl:dr: If I were trying to either (i) train a multitask policy from scratch or (ii) train a sim-to-real policy with domain randomization, I'd want to use PPO with a huge batch size. What (if anything) prevents this in RoboPianist?

  - All of [9, 10, 11, 12, 14, 15, 16] use PPO as the policy optimizer. Why choose DroQ? I haven't seen a comparison of DroQ vs PPO in dexterous manipulation works or the original DroQ paper. DroQ seems to be focused on sample efficiency and in the original paper, it's evaluated in domains with very few samples (~1e5). If it's more suitable than PPO for dexterous manipulation tasks, that would be novel and useful information for practitioners.
  - By my math, 5M samples, with a control frequency of 20Hz, is equivalent to 70 hours of data, and this is collected in 5 hours, yielding a 14x realtime factor. This is 1-2 orders of magnitude less than most recent works in dexterous manipulation, mostly implemented in IsaacGym. Is this because of (a) MuJoCo being a bit slower? (b) the task is more complex involving two higher-dof hands and an articulated piano? (c) or maybe the bottleneck is actually DroQ taking a long time to compute the policy update?

- The benchmark lacks environmental aspects that approximate sim-to-real transfer. Specifically, domain randomization, lag modeling, sensor noise, and the infeasibility of directly observing the key angles. These are worth mentioning in the limitations section when discussing how RoboPianist might transfer to a real robot.

**Robotics Focus:**

Relevant but unlikely to deploy to hardware in near future

**Summary Of Paper:**

Existing simulated tasks for studying dexterous manipulation do not require diverse bimanual behaviors. The authors propose a benchmark consisting of playing 150 piano songs with two Shadow hands. They evaluate this RoboPianist trained with DroQ and find that it outperforms an MPC baseline. Extensive ablations show the impact of various design choices in simulation, reward, and optimization hyperparameters. The 150 songs are shown to have diverse difficulties and are distinct enough that generalizing to a new song or learning multiple songs from scratch is difficult.

**Summary Of Recommendation:**

RoboPianist makes a novel high-dimensional control task accessible to practitioners and thoroughly evaluates a strong baseline for performance. Clarifying the motivation behind atypical algorithm selection and adding an expanded discussion of sim-to-real feasibility will resolve some outstanding questions.

---

> ### Author Response · Authors · 2023-08-06
> **Response to reviewer eFF7**
>
> Thank you for taking the time to review our paper and for providing helpful comments!  We are glad you found the benchmark valuable, that its contributions were significant, and that our evaluations were thorough. We address your concrete questions in our response below; please let us know if you have any remaining concerns.
>
> > tl:dr: If I were trying to either (i) train a multitask policy from scratch or (ii) train a sim-to-real policy with domain randomization, I'd want to use PPO with a huge batch size. What (if anything) prevents this in RoboPianist? All of [9, 10, 11, 12, 14, 15, 16] use PPO as the policy optimizer. Why choose DroQ? I haven't seen a comparison of DroQ vs PPO in dexterous manipulation works or the original DroQ paper. DroQ seems to be focused on sample efficiency and in the original paper, it's evaluated in domains with very few samples (~1e5). If it's more suitable than PPO for dexterous manipulation tasks, that would be novel and useful information for practitioners. By my math, 5M samples, with a control frequency of 20Hz, is equivalent to 70 hours of data, and this is collected in 5 hours, yielding a 14x realtime factor. This is 1-2 orders of magnitude less than most recent works in dexterous manipulation, mostly implemented in IsaacGym. Is this because of (a) MuJoCo being a bit slower? (b) the task is more complex involving two higher-dof hands and an articulated piano? (c) or maybe the bottleneck is actually DroQ taking a long time to compute the policy update?
>
> Thank you for your question. We appreciate the opportunity to clarify our choice of DroQ over PPO as the learning method for RoboPianist. First, during our initial experiments, we tested PPO (sourced from the well-tested [Stable Baselines3](https://stable-baselines3.readthedocs.io/en/master/)) and found that despite utilizing 10x the number of environment samples, PPO generally achieved lower performance and was less stable during training for our tasks as compared to DroQ using near-standard hyperparameters for each. An example learning curve is available on our updated website at [robopianist.github.io](https://robopianist.github.io/).
>
> We additionally observed that the wall clock time for PPO was significantly longer (on the order of days). This poor efficiency can indeed be attributed to the choice of the simulator; PPO is especially well-positioned when using IsaacGym because it can take advantage of thousands of instances on a single GPU (whereas we could spawn only tens of environments at once using MuJoCo). As you pointed out, many works on dexterous manipulation that leverage PPO necessitate the use of a simulator such as IsaacGym, and in principle, it is entirely possible to reimplement this benchmark in IsaacGym and use PPO.
>
> That being said, why did we choose MuJoCo?
>
> 1. Ease of model and task creation. For example, the piano model is generated entirely in code ([see here](https://github.com/robopianist/robopianist-corl23/blob/main/robopianist/models/piano/piano_mjcf.py)), and the piano task is modularly composed of base components (e.g. a piano-only task and a piano with one hand task) using [PyMJCF](https://github.com/deepmind/dm_control/blob/main/dm_control/mjcf/README.md). This allowed us to quickly iterate and make progress on the benchmark in the early stages of the project.
> 2. To leverage the highly-efficient and performant MPC implementation (see [MuJoCo MPC](https://github.com/deepmind/mujoco_mpc)) as a baseline, because it had demonstrated [very strong results](https://youtu.be/Bdx7DuAMB6o?t=138) on the previously-considered-challenging dexterous task of one-handed cube re-orientation [[2](https://arxiv.org/abs/1808.00177), [3](https://arxiv.org/abs/1909.11652), [4](https://arxiv.org/abs/2210.13702)] in simulation.
> 3. Open-source and responsive developers, which was helpful during the creation of the benchmark.
>
> In summary, given our choice of simulator, we found that DroQ–a regularized version of the SAC algorithm which has been used in numerous hard tasks [[5](https://arxiv.org/abs/1812.05905), [6](https://arxiv.org/abs/2304.09834)]–achieved better performance, faster wall clock time, and greater sample efficiency.
>
> > The benchmark lacks environmental aspects that approximate sim-to-real transfer. Specifically, domain randomization, lag modeling, sensor noise, and the infeasibility of directly observing the key angles. These are worth mentioning in the limitations section when discussing how RoboPianist might transfer to a real robot.
>
> We wholeheartedly agree. We’ve added a sim2real transfer section on the website and will include it in the limitation section for the camera-ready version!
>
> Thanks again for your time reviewing our paper and for the insightful questions and comments. We hope that we have addressed your concerns; please let us know if you have any remaining questions.

---

> > ### Author Response · Authors · 2023-08-09
> > **Re: Rebuttal by authors**
> >
> > We again thank the reviewer for the constructive feedback. Please let us know if our response has addressed your concerns about the paper; we are happy to clarify any other concerns/questions that may be remaining.

---

> > ### Author Response · Authors · 2023-08-11
> > **Friendly ping**
> >
> > Just a friendly ping in case this slipped through the cracks. We'd really appreciate a response so we can improve our submission. We appreciate your time and constructive feedback.

---

> > ### Comment · Reviewer_eFF7 · 2023-08-15
> > **Reply from Reviewer eFF7**
> >
> > Thank you for your response. I appreciate the addition of the PPO experiment and sim2real discussion. My takeaway is that DroQ + MuJoCo is an appropriate algorithm + simulator pairing for this work, given its higher performance compared to PPO and qualitatively good behavior in the task. I am still curious how this combination would scale to the sim2real version of the task where domain randomization and partial observations might make the policy search more data-hungry. Since I appreciate the quality of the work and my outstanding questions were addressed in the rebuttal, I maintain my original recommendation for acceptance.

---

### Official Review · Reviewer_Dnam · 2023-07-19

**Confidence:** 5
**Originality:** Good
**Technical Quality:** Good
**Clarity Of Presentation:** Very Good
**Impact:** 4

**Recommendation:**

Weak Accept: I recommend accepting the paper, but will not argue for my recommendation if the majority of other reviewers have a different opinion.

**Review:**

The paper is clear and the approach is sound. The authors address several problems that arise in the development of the agent and propose effective solutions. The contributions are clear and set the basis for interesting future work.

**Quality Of The Limitations Section:**

Limitations are addressed clearly

**Questions For Rebuttal:**

The paper is interesting and well presented. However, the relevance of this work may be increased with some further experiments/corrections:
- as most of the contributions rely on the choices of observations and rewards, the authors should present an ablation study regarding the role of such choices;
- as the MPC results are consistently lower than the model, the authors should also test whether using another backbone model could play an essential role in the agent performance: other exploration and representation techniques may positively impact the performance
- in the abstract the authors state that the produced tracks are "acoustically pleasing", but they do not provide metrics showing the similarity between the track and the correct execution, or any user-based study that demonstrates such property

**Robotics Focus:**

Highly relevant to robotics but no hardware experiments

**Summary Of Paper:**

The authors present an RL based agent that is trained to play piano using bi-manual anthropomorphic hands. The approach uses DroQ as backbone model. The authors contribute by: (1) developing a novel environment; (2) analyzing several choices for observation and reward shaping; (3) proposing a multi-task imitation learning setting for training a single agent able to play multiple songs. This paper may represent a starting point for future research in several domains, such as robotics, machine learning and arts.

**Summary Of Recommendation:**

The paper is clear and the approach is sound. The authors address several problems that arise in the development of the agent and propose effective solutions. The contributions are clear and set the basis for interesting future work.

---

> ### Author Response · Authors · 2023-08-06
> **Response to reviewer Dnam**
>
> Thank you for taking the time to review our paper and for providing helpful comments!  We are glad you found the paper and its contributions clear and the approach sound. You brought up some helpful suggestions which we discuss below. Please let us know if you have any remaining questions or concerns.
>
> > In the abstract the authors state that the produced tracks are "acoustically pleasing", but they do not provide metrics showing the similarity between the track and the correct execution, or any user-based study that demonstrates such property
>
> * We agree that “acoustically pleasing” is subjective. Since we did not conduct a formal human study to evaluate this, we will remove these statements from the paper.
> * That said, to compare the “similarity between the track and the correct execution” we use the F1 score (please see Section 3, “Evaluation Criteria”). The F1 score is a common heuristic accuracy score for various audio information retrieval and signal processing tasks.
>
> > as most of the contributions rely on the choices of observations and rewards, the authors should present an ablation study regarding the role of such choices;
>
> * We detail the specific design choices that were important for successful learning (on top of the basic setup described in Section 3) in Section 4. For these design decisions, we present experimental results to analyze each’s effect in Figure 2. For example, we see that adding an energy penalty to the reward function and adding future goal states to the observation improve performance and reduce variance across seeds. We have updated Figure 2 on the website ([robopianist.github.io](https://robopianist.github.io/)) with an additional reward ablation wherein we remove the fingering reward term, which corresponds to our “human priors” paragraph in Section 4. The effect on performance is substantial—without the fingering information, the agent is never able to make any learning progress. We also note that Section 5.3 analyzes the effect of some additional parameters: control frequency and the lookahead horizon, which indirectly affect the observation space.
> * To summarize, we have already conducted ablations on the reward function and observation space (see Section 4 and section 5.3) and we've just added an additional reward ablation on the website.
>
> > as the MPC results are consistently lower than the model, the authors should also test whether using another backbone model could play an essential role in the agent performance: other exploration and representation techniques may positively impact the performance
>
> Deep RL methods (such as DroQ which we use in RoboPianist) are known to have different strengths and weaknesses when compared to on-the-fly model-based optimization techniques (such as the “Predictive Sampling” method we compare to as our MPC baseline). In regards to our specific choice of deep RL algorithm, we initially tried a widely-used, standard implementation of the popular policy gradient method PPO. We found that even with 10x the number of samples, using the default Stable Baselines3 hyperparameters, PPO was not able to match the performance of the DroQ analog (we have included the learning curves for this experiment on the updated website). While we expect that with sufficient tuning PPO might perform similarly, we chose the backbone algorithm according to its stability and performance with minimal environment-specific tuning. We leave attempts to study different exploration and representation learning techniques for future work.
>
> We will include all updates and clarifications in the final version of the paper. Thanks again for your time reviewing our paper and for the helpful suggestions. We hope that we have addressed your concerns; please let us know if you have any remaining questions.

---

> > ### Author Response · Authors · 2023-08-09
> > **Re: Rebuttal by authors**
> >
> > We again thank the reviewer for the constructive feedback. Please let us know if our response has addressed your concerns about the paper; we are happy to clarify any other concerns/questions that may be remaining.

---

> > ### Author Response · Authors · 2023-08-11
> > **Friendly ping**
> >
> > Just a friendly ping in case this slipped through the cracks. We'd really appreciate a response so we can improve our submission. We appreciate your time and constructive feedback.

---

### Decision · Program_Chairs · 2023-08-30

**Decision:**

Accept (Poster)

**Comment:**

Scores:  Dnam: Weak Accept, eFF7: Weak Accept, 28kq: Weak Accept, 28kq: Weak Accept

Quality:  The paper presents an RL based agent that is trained to play piano in simulation using bi-manual anthropomorphic hands.
It introduces a multi-task imitation learning setting for training a single agent which is able to play multiple songs.
All reviewers have acknowledged that the study is challenging and very well executed.

Clarity: The paper is clear and well-organized

Originality: Good

Significance: The paper presents a novel high-dimensional control task and evaluates a strong baseline for performance, which can serve as a good starting point for future studies.

Cons:

- lack of a user-based study that demonstrates that the agent plays "acoustically pleasing"

Pros:

- novel high-dimensional dexterous control task highly relevant to robotics
- clear contributions with a good baseline for future work
- thorough evaluation of the model, extensive ablation study and good discussion of generalization capabilities